# Adaptations in Mitochondrial Function Induced by Exercise: A Therapeutic Route for Treatment-Resistant Depression

**DOI:** 10.3390/ijms26178697

**Published:** 2025-09-06

**Authors:** Arnulfo Ramos-Jiménez, Mariazel Rubio-Valles, Javier A. Ramos-Hernández, Everardo González-Rodríguez, Verónica Moreno-Brito

**Affiliations:** 1Institute of Biomedical Sciences, Autonomous University of Ciudad Juarez, Ciudad Juarez Campus, Chihuahua 32310, Mexico; 2Faculty of Physical Culture Sciences, Autonomous University of Chihuahua, University Circuit, Campus II, Chihuahua 31125, Mexico; p305510@uach.mx; 3Faculty of Medicine, Autonomous University of Nuevo Leon, University Campus, Monterrey 64460, Mexico; javier.ramoshrn@uanl.edu.mx; 4Faculty of Medicine, Autonomous University of Chihuahua, University Circuit, Campus II, Chihuahua 31125, Mexico; evegonzal@uach.mx (E.G.-R.); vmoreno@uach.mx (V.M.-B.)

**Keywords:** treatment-resistant depression, mood disorders, physical exercise, sedentarism, mitochondrial dynamics, neuroplasticity, neurophysiological disturbances, behavioral stress

## Abstract

Mitochondrial dysfunction is a key factor in the pathophysiology of major depressive disorder (MDD) and treatment-resistant depression (TRD), connecting oxidative stress, neuroinflammation, and reduced neuroplasticity. Physical exercise induces specific mitochondrial changes linked to improvements in mental health. The aim of this paper was to examine emerging evidence regarding the effects of physical exercise on mitochondrial function and treatment-resistant depression, highlighting the clinical importance of the use of mitochondrial biomarkers to personalize exercise prescriptions for patients with depression, particularly those who cannot tolerate standard treatments. Physical exercise improves mitochondrial function, enhances biogenesis and neuroplasticity, and decreases oxidative stress and neuroinflammation. Essential signaling pathways, including brain-derived neurotrophic factor, AMP-activated protein kinase, active peroxisome proliferator-activated receptor-γ coactivator-1α, and Ca^2+^/calmodulin-dependent protein kinase, support these effects. Most studies have concentrated on the impact of low- and moderate-intensity aerobic exercise on general health. However, new evidence suggests that resistance exercise and high-intensity interval training also promote healthy mitochondrial adaptations, although the specific exercise intensity required to achieve this goal remains to be determined. There is strong evidence that exercise is an effective treatment for MDD, particularly for TRD, by promoting specific mitochondrial adaptations. However, key gaps remain in our understanding of the optimal exercise dose and which patient subgroups are most likely to benefit from it (Graphical Abstract).

## 1. Introduction

Major depressive disorder (MDD) is a serious mental health condition characterized by persistent low mood, loss of interest or pleasure in activities, and significant impairment in daily functioning, affecting how a person feels, thinks, and behaves [1]. MDD affects approximately 5% of the adult population worldwide, with 30% of them developing treatment-resistant depression (TRD), which is characterized by an inadequate response to two or more antidepressant treatments or therapies [2]. This means that many people cannot tolerate or do not respond to standard pharmacotherapies, leaving clinicians with few options for treating this health problem. Some critical consequences of MDD and TRD include poor quality of life [3], high healthcare costs, and lost productivity, with annual expenses in the U.S. alone estimated to be between $29 billion and $48 billion [4]. Additionally, there is an increased rate of suicide attempts and self-harming behaviors [5,6]. This situation makes MDD a leading cause of disability worldwide [7]. Nevertheless, physical exercise has been shown to protect against MDD and improve quality of life [8]. This protection is associated with improvements in skeletal muscle and neuronal mitochondrial function [9,10]. Exercise is low-cost, broadly accessible, and can be combined with medications and psychotherapy, offering a route for patients who cannot tolerate higher drug doses or polypharmacy. However, there is no clear guidance on which exercise modality or optimal dose could help to improve mitochondrial dysfunction in TRD. Conversely, prolonged periods of sedentarism have been associated with mitochondrial dysfunction and depressive symptoms [11]. Mitochondrial dysfunction is here defined as the inability of mitochondria to efficiently generate ATP through oxidative phosphorylation in response to cellular energy demands. This dysfunction results from various mitochondrial disturbances, some of which are related to a sedentary lifestyle and, in opposition, physical exercise.

This article highlights and integrates the biochemical mechanisms of neuronal mitochondrial dysfunction caused by systemic stress, proposing that a dose- and personalized physical exercise plan should be integrated into pharmacological and psychological therapies. This integrative plan has significant protective effects to improve mitochondrial function and reduce the symptoms of MDD, particularly in treatment-resistant patients.

## 2. Pathophysiology of Major Depressive Disorder

The biochemical mechanism of depressive disorder has been explained elsewhere [12,13]. These studies strongly suggest that the development of MDD, as well as mood and anxiety disorders, is linked to abnormal brain function [14,15,16,17], hyperactivation of the hypothalamic–pituitary–adrenal axis, severe systemic inflammation, mitochondrial dysfunction, and elevated levels of reactive oxygen species (ROS) [18,19]. Increased levels of inflammatory cytokines and ROS in cerebrospinal fluid interfere with mitochondrial DNA homeostasis, reduce the efficiency of oxidative phosphorylation in the respiratory chain, and hinder mitochondrial biogenesis [18,20,21].

The production of ATP through oxidative phosphorylation is closely linked to the flow of electrons along the inner mitochondrial membrane. However, in individuals with MDD, mitochondrial dysfunction is usually present [22,23], reflected by a proinflammatory profile and elevated levels of cytokines (e.g., interleukin-6, interleukin-8, interleukin-12, IL-1β, and TNF-α) [24,25]. Cytokines damage the electron transport chain (ETC) machinery and promote the release of nitric oxide from microglia, which interferes with cytochrome C oxidase (Complex IV) [24], decreases ATP production, and disrupts neuronal functions vital for mood regulation, such as neurotransmitter release and synaptic plasticity [26]. In animal models of depression, elevated ROS levels are linked to increased rates of mitochondrial DNA (mtDNA) mutations and deletions [27]. Moreover, damaged mtDNA initiates a vicious cycle in which faulty mitochondria produce more ROS, thereby worsening cellular stress and cell dysfunction (Figure 1). Unlike nuclear DNA, mtDNA lacks protective structures, such as histones, which increase its vulnerability to stress damage [28]. A mouse model of human depression and insomnia revealed damage to mitochondrial autophagy, decreased synthesis and secretion of melatonin, and elevated levels of IL-1β, NF-κB, Pink1, and Parkin in the pineal gland [18].

Another pathway of mitochondrial impairment, which has not yet been explored in the context of MDD and may be relevant, involves mutations in the putative kinase 1/Parkin E3 ubiquitin-protein ligase (PINK1/Parkin) genes or their deregulated expression. The PINK1/Parkin pathway is responsible for removing damaged mitochondria in hippocampal neurons [29,30], thus protecting against mitochondrial damage and reducing oxidative stress, both of which are essential for effective neurotrophic signaling. Li et al. (2025) reported increased levels of Pink1/Parkin expression alongside decreased levels of BDNF, Beclin 1, and BCL2 interacting protein 3 expression, a pattern linked to reduced autophagy in damaged pineal gland cells [18].

On the other hand, while selective serotonin reuptake inhibitors (SSRIs) are generally the most common first-line treatments for major depressive disorder (MDD), a significant percentage of patients fail to improve symptoms and instead progress to TRD [31]. Several hypotheses suggest that TRD may originate from severe mitochondrial dysregulation, neuroinflammation, and epigenetic changes that traditional pharmacotherapy does not effectively address [17,32]. Other reasons for pharmacological failure include the fact that each individual reacts differently to treatments; this is why metabolomic and genomic analyses are currently recommended to determine individual sensitivity to treatments. For example, Bhattacharyya et al. (2025) recently reported contrasting differences in the blood concentrations of several neuronal signaling markers in patients with TRD due to the chronic effects of different SSRIs [33].

## 3. Does Chronic Stress Induce Mitochondrial Dysfunction in Patients with Depressive Disorders?

Chronic stress is closely associated with neuronal mitochondrial dysfunction and depressive disorders. Animal models have shown that depression-like behaviors increase ROS levels in neuronal mitochondria and disrupt cellular signaling in the hippocampus and prefrontal cortex [34,35,36,37]. Treatments that restore mitochondrial function, such as mitochondrial transplantation, may help reverse these symptoms (Table 1).

### 3.1. Mitochondrial Dysfunction

Mitochondria are vital organelles involved in various metabolic processes, including energy generation, the biosynthesis of macromolecules, maintaining redox balance, regulating calcium homeostasis, managing cellular waste, and regulating apoptosis. Mitochondrial dysfunction disrupts these processes, leading to pathophysiological issues such as increased ROS production, mitochondrial DNA damage, tissue inflammation, decreased biogenesis, and impaired neuromuscular signaling [38,39], which contribute to the neurophysiological disturbances observed in mood disorders, anxiety, and MDD [22]. This phenomenon occurs in both skeletal muscle and neurons [40,41].

As described above, mitochondrial dysfunction is increasingly recognized as a factor associated with the development of depression. While current research does not prove that mitochondrial dysfunction directly causes depression, there is strong evidence that it contributes to depressive symptoms and that improving mitochondrial function can alleviate these symptoms. Research in both animal models and humans has shown that mitochondrial dysfunction disrupts neurotransmission and neuroplasticity, contributing to depressive symptoms [14,42]. Restoring mitochondrial function, whether through mitochondrial transplantation or exercise, restores mitochondrial function and decreases depressive symptoms [9,35,43].

Associations between chronic psychological stress and mitochondrial alterations have been shown in both human and animal studies [44]. Chronic stress induces morphological changes, including fragmented mitochondria, decreased cristae density, and the downregulation of mitochondrial fusion proteins (FIS1 and OPA1), both in vitro and in rodent models [45,46]. Furthermore, prolonged psychological stress and depression result in chronically elevated cortisol and glucocorticoid levels, which deactivate mineralocorticoid receptors and glucocorticoid receptors, leading to mitochondrial dysfunction and hyperactivity in the hypothalamic–pituitary–adrenal axis [47,48,49].

Another critical process involved in chronic depression is the disturbance of circulating mitochondrial DNA-related microRNAs and mitochondrial calcium deregulation [50]. In patients with depression, four weeks of selective serotonin reuptake inhibitor (SSRI) treatment decreases the levels of circulating mitochondrial DNA and related microRNAs (miR-6068 and miR-4708-3p) and depressive states [50], indicating that mitochondrial damage is linked to depressive states. Chronic stress or inflammatory conditions that lead to calcium overload can disturb the opening of the mitochondrial permeability transition pore and the release of proapoptotic factors such as cytochrome C [51]. Ongoing apoptosis of neuronal cells in the hippocampus and prefrontal cortex, regions vital for mood regulation, may considerably contribute to depression [52]. Additionally, lower brain energy metabolism, as measured by phosphocreatine levels (a marker of mitochondrial function), is correlated with higher depression scores in adolescents [53].

In this context, several therapeutic interventions have been proposed to enhance mitochondrial function and decrease MDD symptoms (Table 2).

### 3.2. Mitochondrial Biogenesis

Mitochondrial biogenesis, the process by which cells increase their mitochondrial mass, is regulated mainly by active peroxisome proliferator-activated receptor-γ coactivator-1α (PGC-1α) [26] and is enhanced by physical exercise [55]. PGC-1α collaborates with nuclear respiratory factor 1 and nuclear respiratory factor 2 to regulate the transcription of nuclear-encoded mitochondrial proteins [56], including Cox10, Cox15, and the β-subunit of ATP synthase. These proteins are utilized for oxidative metabolism, ETC assembly, and mitochondrial ATP synthesis. AMP-activated protein kinase (AMPK) senses the energy status of a cell. It can be activated when there are low levels of ATP, a condition often found in response to exercise [57]. AMPK is activated during exercise and phosphorylates downstream substrates, such as PGC-1α, thereby increasing the activity of genes related to mitochondria and promoting mitochondrial biogenesis [58].

Additionally, the Ca^2+^/calmodulin-dependent protein kinase II pathway is activated by increased intracellular Ca^2+^ during exercise, resulting in the phosphorylation of cAMP response element-binding protein (CREB) and subsequent transcription of BDNF [59]. This signaling improves mitochondrial biogenesis and synaptic plasticity, as well as neuroprotection, which have positive effects on cognitive function through impacts on neurogenesis and the remodeling of synaptic contacts, all of which are essential for managing stress and depression.

### 3.3. Oxidative Stress and Mitochondrial Dynamics

Mitochondria play a central role in both the production of ROS and detoxification; these biochemical mechanisms are well-documented elsewhere. Briefly, during oxidative phosphorylation, the ETC transfers electrons from NADH and FADH_2_ to oxygen, producing H_2_O. A small fraction of these electrons prematurely reduces O_2_ to superoxide (O_2_^−^), which is then enzymatically converted to hydrogen peroxide (H_2_O_2_) and, via metal-catalyzed reactions, to the highly reactive hydroxyl radical (•OH). Oxidative stress occurs when ROS production surpasses the antioxidant defenses, leading to damage of lipids, proteins, and mitochondrial DNA, and disrupting signaling and energy production [60,61]. Mitochondrial ROS mainly originate from complexes I (NADH: ubiquinone oxidoreductase) and III (ubiquinol: cytochrome C oxidoreductase), both of which produce superoxide [60,61,62,63]. Among ROS, the hydroxyl radical (•OH), produced in the mitochondrial matrix, is the most inherently harmful because it reacts almost at diffusion-controlled rates with nearly all biomolecules and lacks specific enzymatic scavengers. It mainly forms from H_2_O_2_ through Fenton and Haber-Weiss reactions involving redox-active iron or copper, causing severe damage at the site of its formation, including mitochondrial DNA and inner-membrane polyunsaturated lipids. Peroxynitrite (ONOO^−^), formed from the near diffusion-limited reaction of superoxide with nitric oxide, is also highly cytotoxic; it oxidizes and nitrates proteins, inactivates enzymes, especially those with iron–sulfur clusters, and initiates lipid peroxidation, further impairing mitochondrial function [64]. By contrast, superoxide itself is short-lived and largely compartmentalized, and H_2_O_2_, although less reactive, becomes dangerous when its local concentration exceeds the capacity of peroxidases [60,61]. In this paper, the importance of ROS lies in its role as a source of mitochondrial pathologies, where MDD and TRD are implicated.

Neuronal cells are highly vulnerable to oxidative damage because they depend on high and efficient energy production by mitochondria. Antioxidant defenses in mammals against the damaging effects of ROS during exercise and for the maintenance of redox homeostasis include the activities of enzymatic antioxidants such as superoxide dismutase (SOD), catalase (CAT), and glutathione peroxidase (GSH-Px) [65].

SOD, CAT, and GSH-Px are the most potent enzymatic antioxidant defenses against oxidative stress and neuronal oxidative damage. The first line of defense includes SOD, which catalyzes the dismutation of superoxide anions to hydrogen peroxide; then, CAT and GSH-Px breakdown hydrogen peroxide into water and molecular oxygen. Table 3 illustrates that mitochondrial dynamics encompass two essential processes: fusion and fission, which are crucial not only for maintaining mitochondrial function but also for preserving morphology, distribution within neurons, and repairing damaged mtDNA. The above-described process is controlled by MFN1 and OPA1, which, through DRP1 and FIS1, regulate this process. This coordinated activity ensures proper mitochondrial function, distribution, and quality control, thereby facilitating optimal neuron health. Protein dysregulation has been noted to be involved in neurodegeneration and impairments in neuronal health [66]. The normalization of mitochondrial dynamics prevents neuronal degeneration and cognitive impairments [38]. Exercise-induced regulation of fusion and fission proteins (dynamin-related proteins) in the direction of control maintains a healthy mitochondrial network [67].

### 3.4. Neuroplasticity and Depression

Neuroplasticity refers to the brain’s ability to adapt to stressors, encode experiences, and recover from physical and metabolic injuries [68]. Neuronal resilience and plasticity depend on proper glial function, particularly that of astrocytes and microglia, which also require intact mitochondrial function for neuronal metabolism and the regulation of the neuroinflammatory stress environment [69,70,71].

Depression and chronic stress are associated with disruptions in neuronal signaling pathways and synaptic plasticity, particularly in brain regions such as the prefrontal cortex and hippocampus [49,72]; where mitochondria play a central role in holding neuroplasticity by supplying the energy (ATP) needed for neurite outgrowth, synapse formation, and long-term potentiation [73]. A high concentration of mitochondria is present in presynaptic terminals, supplying ATP for synaptic vesicle recycling and maintaining the ionic gradients needed for excitability and synaptic transmission [74,75]. Mitochondrial dysfunction caused by chronic stress can drive glial cells toward a proinflammatory state, disrupting synaptic homeostasis and leading to synaptic atrophy, as observed in depressive disorders [76]. Chronic stress impairs mitochondrial energy production in astrocytes, leading to reduced glutamate removal from synapses and increased excitotoxicity, particularly in the prefrontal cortex and hippocampus [77,78,79,80,81]; this ultimately hinders the survival and plasticity of neurons, contributing to the development of depressive symptoms [49]. Rial et al. (2016) reported that in depression, there is a notable decline in astrocyte density and function, accompanied by increased microglial activation in frontolimbic regions, which may contribute to synaptic damage [82].

In contrast, physical exercise significantly enhances synaptic plasticity through various structural and molecular mechanisms, benefiting cognitive functions and facilitating recovery from neurological conditions. Specifically, an 8-week treadmill exercise program increased excitability and synaptic transmission, as well as short- and long-term potentiation, in the hippocampus of 6-month-old APP/PS1 mice. This improvement was correlated with an increase in the number of synaptic structures [83]. In addition, exercise increases synaptic plasticity by promoting the expression of CaMK2a and CYFIP1 through the upregulation of CaMK2a, both of which are involved in dendritic remodeling and synaptic strength [84]. Additionally, the activity of AMPAR following aerobic exercise increases the caveolin-1/VEGF signaling pathway, contributing to the enhancement of synaptic plasticity [85].

Given that altered neuroplasticity and mitochondrial function are connected to the development of depression, identifying new treatment options, such as physical exercise, could serve as an effective intervention to reduce mitochondrial damage and neuroinflammation, thereby restoring neuroplasticity and decreasing mood disorders.

### 3.5. Role of BDNF

BDNF is a neurotrophin essential for neuronal survival, synaptic plasticity, and brain function. It is highly expressed in the brain, particularly in the cortex, hippocampus, basal forebrain, and other regions crucial for learning and memory, and may also function as a protective mechanism against central nervous system diseases, such as depression [86]. Neuronal BDNF levels increase via N-methyl-d-aspartate receptor activation and bind to the tropomyosin receptor kinase B, localized to mitochondria, activating protein kinase A signaling and phosphorylation of proteins Drp1 and Miro-2 [87]. This cascade enhances mitochondrial fusion, trafficking, and content in neurons, leading to increased mitochondrial respiration and ATP production [87].

Decreased BDNF levels, in most brain regions observed during depressive conditions, contribute to the pathogenesis of mood disorders through various interactions with neurotransmitter systems [88]. Conversely, high BDNF levels are strongly correlated with positive outcomes, as this factor promotes neuronal survival, differentiation, and plasticity pathways through the modulation of intracellular calcium and gene transcription related to mitochondrial respiration [87,89].

The synthesis of BDNF is one of the significant effects of exercise, helping in the formation and remodeling of synaptic connections, stimulating the production of new neurons, and reinforcing existing connections by activating genes that promote the growth and stabilization of brain cells. During exercise, increases in β-hydroxybutyrate and lactate stimulate the synthesis of BDNF, as both molecules cross the blood–brain barrier and induce its expression within the brain [90,91]. Animal and in vitro studies have shown that β-hydroxybutyrate increases BDNF expression in hippocampal neurons by activating the cAMP/PKA/p-CREB signaling pathway [92] and through tropomyosin receptor kinase B [91], thereby enhancing neuronal activity. In vitro studies also indicate that lactate promotes hippocampal BDNF expression. During high-intensity interval training (HIIT), blood lactate concentrations increase significantly [93].

### 3.6. Nutrients in Individuals with Depressive Disorders

Vitamins B6 and B12, together with tryptophan, maintain mitochondrial function, and vitamins B6 and B12, together with tryptophan, maintain mitochondrial function and prevent neuronal damage. Vitamin B6 can antagonize mitochondrial dysfunction and oxidative stress in the hippocampus, regardless of stress conditions, by modulating key signaling pathways (p-JNK/Nrf-2/NF-κB) and restoring synaptic protein levels [94]; this could help reduce depressive disorders. Similarly, Didangelos et al. showed that vitamin B12 supplementation in diabetic rats reduces neuronal apoptosis and degeneration, restoring neurotrophic support and enhancing synaptic plasticity [95], which highlights the importance of this nutrient in maintaining mitochondrial health and overall neuronal integrity under metabolic stress conditions. Tryptophan metabolism, particularly through the kynurenine pathway, is closely linked to brain health and mitochondrial function in neural cells [96]. Tryptophan deficiency may reduce the viability of mitochondria in neural cells, specifically through the disruption of the tryptophan-kynurenine pathway [97]. The altered metabolism of tryptophan under conditions of neuroinflammation or stress reroutes more than normal amounts toward kynurenine owing to increased activity of enzymes such as indoleamine 2,3-dioxygenase. This enzyme catalyzes the first rate-limiting step of tryptophan catabolism, which is part of its route to form serotonin and melatonin [35]. This shift is associated with a decrease in mitochondrial membrane potential and ATP production, along with an increase in neurotoxic metabolites, all of which contribute to mitochondrial dysfunction and neuronal damage, as observed in animal models of stress and brain injury. Conversely, tryptophan supplementation has been shown to improve mitochondrial function, increase antioxidant capacity, and reduce inflammation and cell death in animal models subjected to stress [98]. Therefore, a deficiency of these nutrients could result in mitochondrial dysfunction and increased oxidative stress. Although direct evidence of mitochondrial damage from low blood levels of B6, B12, and tryptophan is limited, these results suggest that low blood concentrations of these nutrients may impair mitochondrial function, especially under chronic stress, because they act as neuroprotective agents by reducing oxidative stress, inflammation, and apoptosis, factors often linked to mitochondrial dysfunction in neurodegenerative and metabolic diseases. Overall, these findings emphasize the potential of B vitamins and tryptophan as therapeutic supplements to support mitochondrial health and protect the brain against neurodegeneration and depressive disorders.

## 4. Does Physical Exercise Protect Against Depressive Disorder?

Exercise is an intervention widely associated with various health benefits, including improved cardiovascular function and neuroprotective properties for several psychiatric and neurological disorders [99]. The antidepressant effects of exercise are also well established [100]; however, it has only recently been linked to mitochondrial biogenesis. This relationship becomes more relevant in TRD populations, as the traditional pharmacological approach often presents suboptimal and sometimes unstable relief. Hence, the mitochondria-centered approach is a valuable methodology to optimize the prescription of exercise for mental health-related disorders where conventional treatments do not work.

Regular physical exercise is a potent nondrug therapy capable of preventing and reducing neuronal inflammation through several pathways: lowering the levels of proinflammatory cytokines (e.g., TNF-α and IL-6), increasing the expression of anti-inflammatory cytokines (e.g., IL-10 and IL-35), enhancing antioxidant capacity, and increasing BDNF levels [101,102]. These effects contribute to decreased oxidative damage in neuronal tissues, improved mood and brain health, reduced neuropathic pain, and enhanced cognitive and motor skills. Specifically, exercise increases the levels of neurotrophic factors (e.g., BDNF), which are essential for neuronal survival, development, and plasticity [103]. Furthermore, BDNF promotes homeostatic mitochondrial turnover, particularly by regulating the expression of proteins involved in mitochondrial fusion and fission [104,105]. In this context, endurance exercise increases mitochondrial fusion and fission (autophagy/mitophagy) processes in the brain, thereby increasing mitochondrial turnover and maintaining a healthy cellular environment [105]. As a result, the combined effect of all these pathways decreases neuroinflammation, enhances synaptic plasticity and mitochondrial function, and helps to reduce depression symptoms.

A cross-sectional study found that men with depressive symptoms exhibited significantly lower serum testosterone levels compared to non-depressed controls [106]. Another mouse model study found parallel increases in both testosterone and BDNF following aerobic exercise training [107]. Animal studies indicate that testosterone can modulate BDNF levels in the brain in a dose-specific manner, influencing cognitive and neuroprotective functions [108]. However, a direct association between serum testosterone levels and BDNF in individuals with depressive symptoms has not been established.

### 4.1. Sedentarism and Depression

Sedentary behavior is characterized by extended periods of physical inactivity [11] and is associated with several health problems, including muscle atrophy, neurodegenerative diseases, mitochondrial dysfunction, and depression symptoms. These problems are influenced by several factors, such as age, mobility limitations, sleep impairments, pain, anxiety, and social isolation [109]. Adults who spend 50% or more of their leisure time in sedentary activities experience more frequent depression and anxiety symptoms [110]. Endrighi et al. (2016) reported that two weeks of free-living sedentary time can lead to mood disturbances in healthy adults [111]. Additionally, Schuch et al. (2017) reported that individuals with MDD perform less physical activity and are more involved in sedentary behavior [112].

Systematic physical exercise programs in populations with MDD have been shown to decrease depressive symptoms and enhance overall mental health, and improvements associated with positive changes in mitochondrial function [8,113] suggest that replacing sedentary activities with light, moderate, or vigorous physical activity combined with adequate sleep may significantly reduce symptoms of depression [114].

### 4.2. Sedentarism and Mitochondrial Dysfunction

Sedentarism, or prolonged physical inactivity, has been shown to impair muscle mitochondrial function [115]. Multiple studies in animals and humans have confirmed that a sedentary lifestyle reduces mitochondrial capacity, increases oxidative stress, and impairs metabolic flexibility [79,116]. Figueiredo et al. (2009) reported that, compared with their active counterparts, lifelong sedentary behavior in mice results in a decline in skeletal muscle mitochondrial function, with increased oxidative damage to mitochondrial biomolecules and a greater loss of muscle mass [117]. In rats, a short-term switch from an active to a sedentary routine results in a rapid increase in oxidative stress, a decrease in the activity of antioxidant enzymes (superoxide dismutase, catalase, and glutathione peroxidase), and an increase in the levels of oxidative damage markers (e.g., protein oxidation and 4-hydroxynonenal) in muscle tissue [118].

Studies of in vitro (neuronal cells of a mouse model) sedentary states (such as chronic sleep fragmentation, which often accompanies sedentary lifestyles) have revealed a reduced number of mitochondria, impaired mitochondrial respiratory chain components, and decreased mitochondrial DNA in critical brain regions for cognition. These changes are associated with disrupted mitochondrial biogenesis signaling pathways and worsened cognitive function [119]. Furthermore, in a mouse model of mitochondrial disorders, sedentary behavior was associated with pronounced cerebellar dysfunction and mitochondrial deficiency [41] (Figure 2).

### 4.3. Effects of Different Types of Exercise on Mitochondrial Function and Depression

Physical exercise is increasingly recognized for its role in enhancing both mitochondrial function and mental health, particularly in the treatment of depression. Different types of exercise, as mentioned below, influence biological and psychological outcomes in distinct ways.

#### 4.3.1. Aerobic Exercise

Aerobic exercise training (such as running, cycling, or swimming at moderate intensity) is well documented to enhance brain mitochondrial function and mitochondrial oxidative capacity. Studies in a mouse model of Alzheimer’s disease have shown that aerobic exercise upregulates CD38 expression in astrocytes, facilitating the CD38-mediated transfer of healthy mitochondria from astrocytes to neurons [9]. Additionally, it increases PGC-1α, SIRT1, and the expression of citrate synthase, as well as microRNA, within hippocampal astrocytes and across most brain regions, promoting the transfer of healthy mitochondria from astrocytes to neurons and regulating mitochondrial proteostasis, which in turn increases brain mitochondrial biogenesis [10]. In old Wistar rats, aerobic exercise increased the expression of the OPA1 gene in the hippocampus, reduced the expression of the Drp1 gene, modulated mitochondrial fusion and fission processes, and improved spatial learning and memory performance [120]. Fernández et al. (2020) reported that combined exercise training (aerobic and resistance) increases mitochondrial complex V activity in the brains of a mouse model [41]; however, animal studies are not always transferable to humans. While there are conserved elements of mitochondrial metabolism across species [121], significant differences exist, particularly in how metabolic dysfunctions manifest and are regulated [122].

#### 4.3.2. Resistance Exercise

Resistance training has long been a subject of discussion regarding mitochondrial adaptations. Animal studies (in rats with sporadic inclusion body myositis) have shown that resistance training reduces damage and skeletal muscle atrophy, increases mitochondrial biogenesis, and decreases amyloid-beta protein (Aβ) accumulation [123,124]. However, human studies have shown little to no effect on mitochondrial function [125]. Twelve weeks of resistance training increased muscle strength and muscle oxidative capacity, increasing the proportion of neural cell adhesion molecule-positive satellite cells and restoring normal mitochondrial function in patients with sporadic mitochondrial DNA mutations in skeletal muscle [126]. A 21-day resistance training program following 10 days of muscle disuse increases muscle mass, function, mitochondrial activity, and biogenesis [127]. Twelve weeks of resistance exercise training increased the mitochondrial respiratory capacity and muscle strength without inducing mitochondrial biogenesis in young, healthy men [128]. Six weeks of high-volume resistance training decreased citrate synthase activity by 24% and reduced the concentrations of actin and myosin proteins in the muscle fibers of college resistance-trained individuals [129]. Thus, the findings regarding the benefits of resistance training on mitochondrial function are inconsistent.

#### 4.3.3. High-Intensity Interval Training

HIIT involves short bouts of near-maximal or maximal effort interspersed with rest or low-intensity exercise periods. HIIT increased lactate levels and promoted hippocampal MCT1/4 and BDNF expression. HIIT can induce skeletal mitochondrial adaptations (increased mitochondrial content, citrate synthase, and cytochrome C oxidase in skeletal muscle) [130,131,132]. HIIT stimulates BDNF expression in the hippocampus, enhancing brain function in a mouse model [93]. One week of HIIT increased BDNF, doublecortin, and voltage-dependent anion-selective channel protein 2 in a mouse model. Nevertheless, it decreased mitochondrial superoxide dismutase 2 content in the hippocampus, with no changes in redox status [133].

The high-energy stress associated with HIIT strongly activates either AMPK or p38 MAPK [134]. Both kinases converge on PGC-1α, leading to increased mitochondrial biogenesis. Studies on rodent models and humans have repeatedly shown that HIIT interventions increase the activity of citrate synthase, an important marker of mitochondrial content [135]. Although HIIT is unlikely to be available to all patients, the significant mitochondrial adaptations promise to represent a time-efficient strategy for augmenting the current treatment landscape for mental health (Figure 3).

### 4.4. Duration, Frequency, and Intensity of Exercise

As mentioned above, moderate- and high-intensity physical exercise induces significant changes in mitochondrial metabolism; however, there is no single type of exercise, intensity, volume, or frequency that works for everyone, as these factors vary depending on the disease and particular needs [136]. Nevertheless, current epidemiological, systematic, and meta-analyses indicate that the effectiveness of physical exercise as a treatment for depression depends on the duration and intensity of the exercise programs. Moderate- and high-intensity exercise interventions lasting more than 8 weeks, with exercise sessions of ≥150 min per week or more, are most effective, reducing depression symptoms by 20% to 60% [137,138,139].

Additionally, factors such as age, baseline fitness, and the presence of chronic diseases can influence responses to exercise [140]. For cases of anxiety and depression, exercise prescriptions must consider the psychological and clinical aspects of the patient, the environment, and personal preferences, which could be crucial in maximizing the antidepressant and anxiolytic benefits of exercise. Furthermore, patients with TRD may experience motivational impediments and have a limited active range of motion due to long-standing depression symptoms. Therefore, structured, progressive exercise interventions are recommended to focus on adherence, enjoyment, and psychological support. Future studies should aim to identify the minimum effective dose that improves mitochondrial function in these patients.

## 5. Summary

This review examines the biochemical and clinical evidence that exercise-induced mitochondrial adaptation can reduce the symptoms of MDD, particularly in treatment-resistant patients. Such evidence may help explain the consistent link between oxidative stress, systemic inflammation, impaired mitochondrial function, chronic psychological stress, and MDD. All these factors impair neuroplasticity and worsen psychiatric prognosis. By focusing on mitochondrial dysfunction, this review reinterprets TRD as a disorder with complex metabolic and inflammatory roots rather than just a neurochemical imbalance, offering new targets for the treatment and recovery of those with chronic depression whose quality of life has been severely affected. The potential for exercise to modulate neuronal metabolism, especially mitochondrial function, in treating psychiatric disorders is promising as a therapeutic approach for MDD; this is especially important given the diversity of depression and the varying ways in which treatment resistance presents. Furthermore, physical exercise is a relatively affordable and accessible option for many communities and can be prescribed alongside drug therapy for patients with TRD.

Mitochondrial biogenesis and antioxidant defenses are promoted by physical exercise through key signaling pathways, including BDNF, AMPK, PGC-1α, and CaMK, which act to counteract the biochemical and cellular insults of stress and depression. Interim studies and clinical trials involving populations with TRD have shown that physical exercise lessens depression symptoms in these patients, while also enhancing mitochondrial function, increasing energy capacity, and boosting antioxidant enzyme activity.

Aerobic, resistance, or high-intensity interval training interventions are effective methods to improve MDD and TRD; however, because of the wide variety of study approaches and different types of physical exercises used, we only have general recommendations and no precise data on the ideal intensity, volume, or type of exercise for treating this condition. As a result, each case remains unique. Considering the evidence presented here, we propose that exercise should be viewed not only as an adjunct but also as a fundamental therapeutic principle for patients with TRD owing to its extensive metabolic, anti-inflammatory, and brain health benefits. Exercise may have a synergistic effect when combined with current pharmacotherapies and psychotherapies. Physicians should encourage their patients to incorporate regular exercise into their daily routines. However, patients with TRD often have lower motivation because of the disease itself. Hence, such structured programs will encompass behavioral support, social interaction, and regular follow-up to sustain adherence and realize maximal long-term benefits.

Future research should include more extensive, well-controlled randomized controlled trials involving treatment-resistant depression patients, incorporating clearly defined mitochondrial biomarkers and standardized exercise protocols to determine causality and refine exercise dosage. While animal models have proven useful in identifying exercise-mediated adaptations, their relevance to humans remains limited. Therefore, there is a strong need for direct studies in humans using either muscle biopsies or peripheral blood mononuclear cell samples. Additionally, innovative noninvasive or minimally invasive methods to assess central mitochondrial function should be developed and thoroughly validated to help identify early disease stages, assess severity, and monitor treatment effectiveness.

In summary, existing evidence indicates that exercise-induced mitochondrial adaptations are promising but underused therapeutic targets in TRD. Future research connecting basic science with clinical practice can help identify the best ways to harness the benefits of exercise to improve the lives of those with treatment-resistant depression.

## Figures and Tables

**Figure 1 ijms-26-08697-f001:**
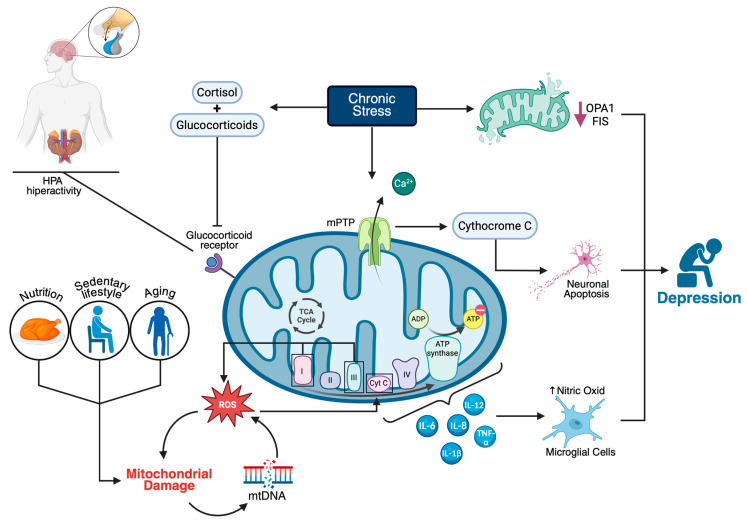
Mitochondrial dysfunction as a central mechanism in the pathophysiology of major depressive disorder (MDD). Chronic stress activates the hypothalamic–pituitary–adrenal (HPA) axis, leading to increased glucocorticoid and cortisol levels, which, through glucocorticoid receptor signaling, contribute to mitochondrial dysfunction. This process is exacerbated by factors such as aging, a sedentary lifestyle, and poor nutrition. Mitochondrial impairment is characterized by increased reactive oxygen species (ROS) in complexes I and III, damage to mitochondrial DNA (mtDNA), altered ATP synthesis, and elevated levels of proinflammatory cytokines (e.g., IL-6, IL-8, IL-12, IL-1β, TNF-α). Chronic stress reduces mitochondrial fusion proteins (OPA1 and FIS), promotes calcium overload and mitochondrial permeability transition pore (mPTP) opening, leading to cytochrome c release and neuronal apoptosis. These cellular alterations disrupt neurotransmission and neuroplasticity, ultimately contributing to the development of depressive symptoms.

**Figure 2 ijms-26-08697-f002:**
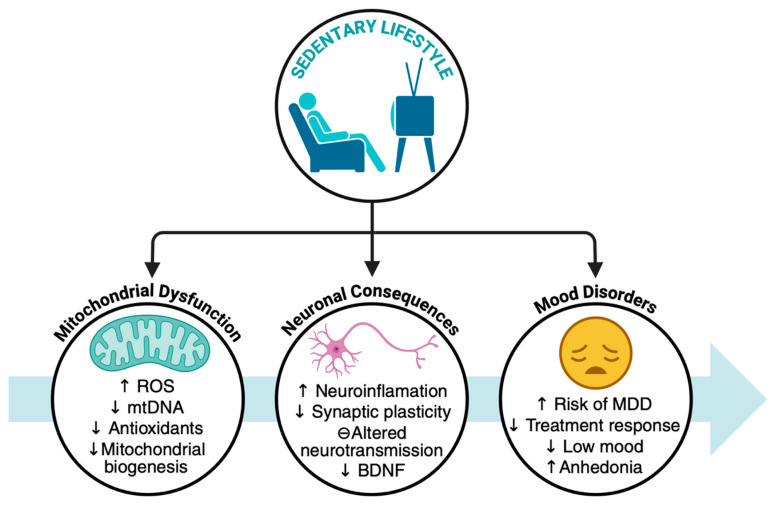
Sedentary lifestyle as a contributing factor to mitochondrial dysfunction, neuronal alteration, and mood disorders. BDNF: Brain-derived neurotrophic factor, MDD: Major depressive disorder, mtDNA: Mitochondrial DNA, ROS: Reactive oxygen species. The arrows mean increase (↑) and decrease (↓).

**Figure 3 ijms-26-08697-f003:**
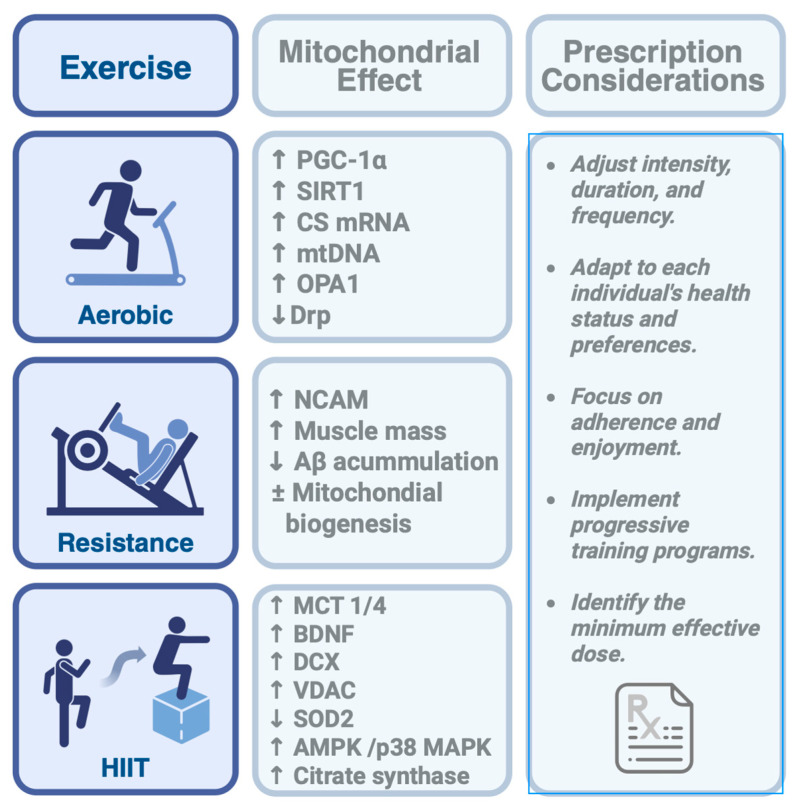
Mitochondrial adaptations and prescription considerations across different types of physical exercise. Aerobic, resistance, and high-intensity interval training (HIIT) each promote specific mitochondrial responses. Aβ: amyloid-beta protein, AMPK: AMP-activated protein kinase, BDNF: Brain-derived neurotrophic factor, CS mRNA: Citrate synthase microRNA, DCX: Doublecortin, MCT: Monocarboxylate transporters, mtDNA: Mitochondrial DNA, NCAM: Neural cell adhesion molecule, OPA1: Optic Atrophy 1, PGC-1α: Active peroxisome proliferator-activated receptor-γ coactivator-1α, p38 MAPK: p38 mitogen-activated protein kinase, SIRT1: Silent Information Regulator T1, SOD2: Superoxide dismutase 2, VDAC: Voltage-dependent anion-selective channel protein 2. The arrows mean increase (↑) and decrease (↓).

**Table 1 ijms-26-08697-t001:** Chronic Stress, Mitochondrial Dysfunction, and Depression-like Behaviors.

Model	Stress Factor	Intervention	Outcome	Citation
Male Wistar rats	Chronic behaviors stress	Mitochondrial transplantation	Restored mitochondrial function and increased ATP	[34,35,36]
Sprague–Dawley rats	Maternal separation	Antidepressants	Restores mitochondrial function and increases ATP	[34]
Adolescent cynomolgus monkeys	Chronic unpredictable mild stress	Chronic stress	Increased metabolic dysfunction and depression	[37]

**Table 2 ijms-26-08697-t002:** Proposed therapeutic interventions to enhance mitochondrial function and reduce depressive disorders or their severity.

Intervention	Model	Effect on Mitochondria	Effects on Depression-Like Behaviors	Citations
Mitochondrial transplantation	Murine	Restores function, increases ATP	Reduces depressive symptoms	[35,36]
Physical exercise	Murine	Normalizes mitochondrial activity	Improves mood, reduces symptoms	[43,54]
Herbal treatments (e.g., Sinisan)	Murine	Improves mitochondrial function	Alleviates depressive behaviors	[34]
Mirtazapine, paroxetine, or sertraline	Human	Improves mitochondrial function	Remission of MDD	[50]
Creatine supplementation	Human	Increases brain energy stores	Correlates with symptom reduction	[53]

**Table 3 ijms-26-08697-t003:** Mitochondrial dynamics of MFN1, OPA1, and DRP1.

Mitofusin 1 (MFN1) is found in the outer mitochondrial membrane and facilitates the fusion of neighboring mitochondria by connecting their outer membranes. It collaborates with MFN2 to preserve the integrity and function of the mitochondrial network. MFN1 is vital for maintaining connected mitochondria in neurons, which is crucial for energy distribution and calcium buffering in axons and dendrites.
Optic Atrophy 1 (OPA1) is a GTPase located in the inner mitochondrial membrane, responsible for fusion within the inner mitochondrial membrane (IMM) and cristae remodeling. It plays a key role in protecting mitochondrial structure and bioenergetics. In neurons, OPA1 is essential for synaptic function and neuroprotection. Abnormal regulation of OPA1 activity has been associated with optic atrophy and Parkinson’s disease.
Dynamin-Related Protein 1 (DRP1) is a GTPase that is vital for mitochondrial fission through constricting and dividing mitochondria at specific sites on the outer mitochondrial membrane. DRP1 is critical for distributing mitochondria in neurons, especially in axons and synapses, where energy demands are high. However, excessive activity of this protein can cause mitochondrial fragmentation, synaptic issues, and neurodegeneration, as observed in Alzheimer’s and Parkinson’s diseases.
Fission 1 Protein (FIS1) collaborates with DRP1 to promote mitochondrial fission. Its precise role in humans is debated, but it is believed to act as a receptor recruiting DRP1. In neurons, FIS1 plays a role in regulating mitochondrial size and distribution. Abnormal regulation of FIS1 is linked to mitochondrial dysfunction and cognitive decline in neurodegenerative diseases.

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
