# Peer review of "Adaptations in Mitochondrial Function Induced by Exercise: A Therapeutic Route for Treatment-Resistant Depression"

_ijms, 2025, doi:10.3390/ijms26178697_

Round 1
Reviewer 1 Report
Comments and Suggestions for Authors
This manuscript introduced the important roles of mitochondria in depression. Physical exercise represents a potential therapy for depression through mitochondria adaptation. The manuscript summarized the underlying mechanisms by which exercise improves mitochondrial functions and the mental health. However, some concerns remain.
- Part 3 could be better organized. Mitochondrial homeostasis encompasses multiple aspects, including energy supply (ATP generation), mitochondrial quantity and structure mitophagy, etc., in addition to mitochondrial biogenesis and dynamics. Furthermore, “3.2 Neuroplasticity and depression” should be relocated before “3.5 Role of BDNF”.
- Based on physical exercise, practical suggestions should be provided for the prevention, detection, and treatment of depression.
Author Response
The authors greatly appreciate the reviewers' rigorous observations, comments, and suggestions. It undoubtedly helps ensure that our manuscripts are publishable with high quality. We respond to them promptly below.
Reviewer 1:
This manuscript introduced the important roles of mitochondria in depression. Physical exercise represents a potential therapy for depression through mitochondria adaptation. The manuscript summarized the underlying mechanisms by which exercise improves mitochondrial functions and the mental health. However, some concerns remain.
- Part 3 could be better organized. Mitochondrial homeostasis encompasses multiple aspects, including energy supply (ATP generation), mitochondrial quantity and structure mitophagy, etc., in addition to mitochondrial biogenesis and dynamics.
R: It was better organized. Lines 139-140.
- Furthermore, “3.2 Neuroplasticity and depression” should be relocated before “3.5 Role of BDNF”.
R: It was relocated. Lines 238-274.
- Based on physical exercise, practical suggestions should be provided for the prevention, detection, and treatment of depression.
R: Additional information on the implications of physical exercise was provided, along with general suggestions on intensity, volume, and frequency for individuals with MDD. Lines 474-482, 514-518.

Reviewer 2 Report
Comments and Suggestions for Authors
Very well conducted review. Could you include a section about different volumes/intensities of training? Also the role of testosterone in the mitochondrial remodeling and anti-inflammatory properties could be of great interest. I listed a study for your reference.
Torabi, Farnaz, and Reza Ahmadi. "The Effect of Exercise Volume on Depressive-Related Behaviors and Levels of Brain-Derived Neurotrophic Factor and Serum Testosterone Levels." International Journal of Sport Studies for Health 7.3 (2024).
Author Response
The authors greatly appreciate the reviewers' rigorous observations, comments, and suggestions. It undoubtedly helps ensure that our manuscripts are publishable with high quality. We respond to them promptly below.
Very well conducted review. Could you include a section about different volumes/intensities of training?
R: Additional information was provided on the implications of physical exercise, along with general suggestions on intensity, volume, and frequency, for individuals with MDD. Lines 474-482, 514-518
Also, the role of testosterone in the mitochondrial remodeling and anti-inflammatory properties could be of great interest. I listed a study for your reference.
R: Additional information was provided about the implications of physical exercise, along with general suggestions on intensity, volume, and frequency, for individuals with MDD. Lines 359-365.

Reviewer 3 Report
Comments and Suggestions for Authors
Adaptations in Mitochondrial Function Induced by Exercise: A Therapeutic Route for Treatment-Resistant Depression
The authors have done significant work, and the manuscript they have submitted undoubtedly has scientific and practical value. The manuscript is written in an appropriate way. However, there are a number of suggestions that the authors should take into account, as well as there are questions and recommendations, which are presented below:
Abstract
- Line 26. "BDNF, AMPK, PGC-1α, and CaMK". The abbreviation should be deciphered. Since BDNF, AMPK, PGC-1α, and CaMK do not appear in the abstract text, these abbreviations do not need to be included in the abstract.
- Line 29. The abbreviation "(HIIT)" should be removed. The abbreviation is not needed in the abstract since there are no other references to HIIT in the abstract.
- Line 34. Is there a graphic abstract? I haven't seen one. If there isn't one, delete the phrase "(Graphical Abstract)".
Introduction
- It is necessary to enhance the novelty and relevance of the problem being studied.
Pathophysiology of major depressive disorder
- Line 60-61. "disorders, is associated with abnormal brain function [14–17], severe mitochondrial dysfunction, systemic inflammation, high levels of reactive oxygen species (ROS), and hyperactivation of the hypothalamic‒pituitary‒adrenal (HPA) axis". Perhaps the order of the phrases in this sentence should be changed. For example "abnormal brain function, hyperactivation of the hypothalamic‒pituitary‒adrenal (HPA) axis severe, systemic inflammation, mitochondrial dysfunction and high levels of reactive oxygen species (ROS). "
- Line 64. "reactive oxygen species (ROS)". Remove either "reactive oxygen species" or the abbreviation "(ROS)". This abbreviation was previously deciphered (Line 61).
- Line 64. "decrease the efficiency of the respiratory chain". Perhaps you meant that they decrease the efficiency of oxidative phosphorylation of the respiratory chain?
- Line 67. "flow of electrons along the inner mitochondrial membrane". Electron flow occurs through the ETC of mitochondria in the inner mitochondrial membrane. It is necessary to add "ETC".
- Line 70. The abbreviation "ETC" should be expanded.
- Figure 1. It is not quite clear what you wanted to show in the figure. Why is the arrow from the ROS directed to the respiratory chain? Or is it directed to cytochrome c? Do ROS formed in the mitochondrial respiratory chain affect the respiratory chain? Probably, it is necessary to note the main sources of ROS formation in the mitochondrial respiratory chain. Under normal conditions, ROS are formed in the respiratory chain, the functions of which are very important. Please think about this Figure 1.
- Line 101. The abbreviation "(BNIP3)" should be removed. This abbreviation does not appear in the text of the manuscript.
- Line 104. Earlier in the introduction there was already a definition of the abbreviation "MDD". Remove either "major depressive disorder" or "(MDD) ".
Does chronic stress induce mitochondrial dysfunction in patients with depressive disorders?
- Line 126. Either "reactive oxygen species" or "ROS" should be removed.
- Line 126-127. "as increased reactive oxygen species (ROS) production, tissue inflammation, decreased biogenesis, mitochondrial DNA damage, impaired neuromuscular signaling, and reduced". Perhaps the order of the phrases in this sentence should be changed. For example, "as increased reactive oxygen species (ROS) production, mitochondrial DNA damage, tissue inflammation, decreased biogenesis, impaired neuromuscular signaling, and reduced".
- Line 146. I recommend removing the abbreviations "(MRs)" and "(GRs)". There are no further references to these abbreviations in the text.
- Line 147. I recommend removing the abbreviation "(HPA)".
- Line 155. I recommend removing the abbreviation "(PTP)".
- Line 203. I recommend removing the abbreviation "(NRF1)" and "(NRF2)".
- Line 205-206. Either " electron transport chain" or "(ETC)" should be removed.
- Line 211. I recommend removing the abbreviation "(CaMKII)".
- Line 218. I recommend to include the information about oxidative stress and the formation of ROS by mitochondria in this subsection. Describe the main sources of ROS formation. Where ROS are formed, in which complexes of the respiratory chain, and which forms of ROS are the most dangerous.
- Line 256. The abbreviation " HIIT" should be expanded.
- Line 274. I recommend removing the abbreviation "IDO".
Does physical exercise protect against depressive disorder?
- Line 304. Either "brain-derived neurotrophic factor" or "(BDNF)" should be removed.
- Line 307. Either "brain-derived neurotrophic factor" or "[BDNF]" should be removed.
- Line 339-342. "In rats, a short-term switch from an active to a sedentary routine results in a rapid increase in oxidative stress, muscle atrophy, decreased activity of antioxidant enzymes (superoxide dismutase, catalase, and glutathi-one peroxidase) and increased levels of oxidative damage markers (protein oxidation and 4-hydroxynonenal) in muscle tissue [109]. " Do rapid increases in oxidative stress, decreased activity of antioxidant enzymes (superoxide dismutase, catalase, and glutathione peroxidase) and increased levels of oxidative damage markers (protein oxidation and 4-hydroxynonenal) in muscle tissue lead to muscle atrophy? I recommend removing "muscle atrophy" from this sentence or changing the sentence. This sentence lists processes occurring at the cell level and between them "muscle atrophy".
- Line 363. Did you mean the oxidative capacity of mitochondria? Please indicate whose oxidative capacity exactly.
- Line 367. I recommend removing the abbreviation "(mRNA)".
- Line 385. Please indicate what oxidative capacity increased as a result of training. Did the oxidative capacity of muscles increase?
- Line 387. I recommend removing the abbreviation "(NCAM)".
- Line 400-401. "cytochrome c oxidase II and IV activity, and mitochondrial content) " What did you mean? Cytochrome c oxidase is complex IV of the respiratory chain. What does II mean? What mitochondrial protein contents were increased? Please edit this sentence.
- Line 400. Please write the same way - "cytochrome c oxidase"- Line 400, "cytochrome C oxidase"- Line 71.
- Line 403. Either "brain-derived neurotrophic factor" or "(BDNF)" should be removed. I recommend removing the abbreviation "(DCX)".
- Line 404. I recommend removing the abbreviation "(VDAC)".
- Line 405. Please clarify, did you observe an increase in protein content or expression of SOD2?
- Line 407. Either "high-intensity interval training" or "(HIIT)" should be removed.
- Line 415. The abbreviation "VDAC" with its explanation should be added to the description of Figure 3.
Summary
- Line 474. Either "treatment-resistant depression high-intensity interval training" or "(TRD)" should be removed.
- Line 475. I recommend removing the abbreviation "(PBMC)".
- Line 480-481. Either "treatment- resistant depression high-intensity interval training" or "(TRD)" should be removed.
Abbreviations should be expanded in the abstract, figure captions, and once in the text of the article. They do not have to be cited and expanded in each chapter of the article.
References
Please check that the references are formatted according to the formatting rules. Lines 653, 705, 715, 724, 737, 764, 772, 839, 847, etc. Somewhere the journal titles are given in full, somewhere - in abbreviated form. The journal titles and year are not highlighted (For example, Lines 563, 565, etc).
Please check that the references are formatted in accordance with the journal rules:
Journal Articles:
1. Author 1, A.B.; Author 2, C.D. Title of the article. Abbreviated Journal Name Year, Volume, page range.
- Books and Book Chapters:
2. Author 1, A.; Author 2, B. Book Title, 3rd ed.; Publisher: Publisher Location, Country, Year; pp. 154–196.
3. Author 1, A.; Author 2, B. Title of the chapter. In Book Title, 2nd ed.; Editor 1, A., Editor 2, B., Eds.; Publisher: Publisher Location, Country, Year; Volume 3, pp. 154–196. - Unpublished materials intended for publication:
4. Author 1, A.B.; Author 2, C. Title of Unpublished Work (optional). Correspondence Affiliation, City, State, Country. year, status (manuscript in preparation; to be submitted).
5. Author 1, A.B.; Author 2, C. Title of Unpublished Work. Abbreviated Journal Name year, phrase indicating stage of publication (submitted; accepted; in press). - Unpublished materials not intended for publication:
6. Author 1, A.B. (Affiliation, City, State, Country); Author 2, C. (Affiliation, City, State, Country). Phase describing the material, year. (phase: Personal communication; Private communication; Unpublished work; etc.) - Conference Proceedings:
7. Author 1, A.B.; Author 2, C.D.; Author 3, E.F. Title of Presentation. In Title of the Collected Work (if available), Proceedings of the Name of the Conference, Location of Conference, Country, Date of Conference; Editor 1, Editor 2, Eds. (if available); Publisher: City, Country, Year (if available); Abstract Number (optional), Pagination (optional). - Thesis:
8. Author 1, A.B. Title of Thesis. Level of Thesis, Degree-Granting University, Location of University, Date of Completion. - Websites:
9. Title of Site. Available online: URL (accessed on Day Month Year).
Unlike published works, websites may change over time or disappear, so we encourage you create an archive of the cited website using a service such as WebCite. Archived websites should be cited using the link provided as follows:
10. Title of Site. URL (archived on Day Month Year).
Author Response
The authors greatly appreciate the reviewers' rigorous observations, comments, and suggestions. It undoubtedly helps ensure that our manuscripts are publishable with high quality. We respond to them promptly below.
Reviewer 3:
Adaptations in Mitochondrial Function Induced by Exercise: A Therapeutic Route for Treatment-Resistant Depression
The authors have done significant work, and the manuscript they have submitted undoubtedly has scientific and practical value. The manuscript is written in an appropriate way. However, there are a number of suggestions that the authors should take into account, as well as there are questions and recommendations, which are presented below:
Abstract
- Line 26. "BDNF, AMPK, PGC-1α, and CaMK". The abbreviation should be deciphered. Since BDNF, AMPK, PGC-1α, and CaMK do not appear in the abstract text, these abbreviations do not need to be included in the abstract.
R: All inappropriate abbreviations were removed throughout the text.
- Line 29. The abbreviation "(HIIT)" should be removed. The abbreviation is not needed in the abstract since there are no other references to HIIT in the abstract.
R: All inappropriate abbreviations were removed throughout the text.
- Line 34. Is there a graphic abstract? I haven't seen one. If there isn't one, delete the phrase "(Graphical Abstract)".
R: Since the first submission, the graphic abstract has been added.
Introduction
- It is necessary to enhance the novelty and relevance of the problem being studied.
R: Following your suggestions, this information has been added: Lines 46-48, 55-58, and 60-69.
Pathophysiology of major depressive disorder
- Line 60-61. "disorders, is associated with abnormal brain function [14–17], severe mitochondrial dysfunction, systemic inflammation, high levels of reactive oxygen species (ROS), and hyperactivation of the hypothalamic‒pituitary‒adrenal (HPA) axis". Perhaps the order of the phrases in this sentence should be changed. For example "abnormal brain function, hyperactivation of the hypothalamic‒pituitary‒adrenal (HPA) axis severe, systemic inflammation, mitochondrial dysfunction and high levels of reactive oxygen species (ROS). "
R: It was corrected. Lines 73-75
- Line 64. "reactive oxygen species (ROS)". Remove either "reactive oxygen species" or the abbreviation "(ROS)". This abbreviation was previously deciphered (Line 61).
R: All inappropriate abbreviations were removed throughout the text.
- Line 64. "decrease the efficiency of the respiratory chain". Perhaps you meant that they decrease the efficiency of oxidative phosphorylation of the respiratory chain?
R: It was corrected. Line 77
- Line 67. "flow of electrons along the inner mitochondrial membrane". Electron flow occurs through the ETC of mitochondria in the inner mitochondrial membrane. It is necessary to add "ETC".
R: All inappropriate abbreviations were removed throughout the text.
- Line 70. The abbreviation "ETC" should be expanded.
R: It was added. Line 83
- Figure 1. It is not quite clear what you wanted to show in the figure. Why is the arrow from the ROS directed to the respiratory chain? Or is it directed to cytochrome c? Do ROS formed in the mitochondrial respiratory chain affect the respiratory chain? Probably, it is necessary to note the main sources of ROS formation in the mitochondrial respiratory chain. Under normal conditions, ROS are formed in the respiratory chain, the functions of which are very important. Please think about this Figure 1.
R: The ROS production process was clarified both in the text (Lines 194-216) and in Figure 1.
- Line 101. The abbreviation "(BNIP3)" should be removed. This abbreviation does not appear in the text of the manuscript.
R: All inappropriate abbreviations were removed throughout the text.
- Line 104. Earlier in the introduction there was already a definition of the abbreviation "MDD". Remove either "major depressive disorder" or "(MDD) ".
R: All inappropriate abbreviations were removed throughout the text.
Does chronic stress induce mitochondrial dysfunction in patients with depressive disorders?
- Line 126. Either "reactive oxygen species" or "ROS" should be removed.
R: All inappropriate abbreviations were removed throughout the text.
- Line 126-127. "as increased reactive oxygen species (ROS) production, tissue inflammation, decreased biogenesis, mitochondrial DNA damage, impaired neuromuscular signaling, and reduced". Perhaps the order of the phrases in this sentence should be changed. For example, "as increased reactive oxygen species (ROS) production, mitochondrial DNA damage, tissue inflammation, decreased biogenesis, impaired neuromuscular signaling, and reduced".
R: It was corrected. Lines 139-140
- Line 146. I recommend removing the abbreviations "(MRs)" and "(GRs)". There are no further references to these abbreviations in the text.
R: All inappropriate abbreviations were removed throughout the text.
- Line 147. I recommend removing the abbreviation "(HPA)".
R: All inappropriate abbreviations were removed throughout the text.
- Line 155. I recommend removing the abbreviation "(PTP)".
R: All inappropriate abbreviations were removed throughout the text.
- Line 203. I recommend removing the abbreviation "(NRF1)" and "(NRF2)".
R: All inappropriate abbreviations were removed throughout the text.
- Line 205-206. Either " electron transport chain" or "(ETC)" should be removed.
R: All inappropriate abbreviations were removed throughout the text.
- Line 211. I recommend removing the abbreviation "(CaMKII)".
R: All inappropriate abbreviations were removed throughout the text.
- Line 218. I recommend to include the information about oxidative stress and the formation of ROS by mitochondria in this subsection. Describe the main sources of ROS formation. Where ROS are formed, in which complexes of the respiratory chain, and which forms of ROS are the most dangerous.
R: The ROS production process was clarified: Lines 194-216
- Line 256. The abbreviation " HIIT" should be expanded.
R: All inappropriate abbreviations were removed throughout the text.
- Line 274. I recommend removing the abbreviation "IDO".
R: All inappropriate abbreviations were removed throughout the text.
Does physical exercise protect against depressive disorder?
- Line 304. Either "brain-derived neurotrophic factor" or "(BDNF)" should be removed.
R: All inappropriate abbreviations were removed throughout the text.
- Line 307. Either "brain-derived neurotrophic factor" or "[BDNF]" should be removed.
R: All inappropriate abbreviations were removed throughout the text.
- Line 339-342. "In rats, a short-term switch from an active to a sedentary routine results in a rapid increase in oxidative stress, muscle atrophy, decreased activity of antioxidant enzymes (superoxide dismutase, catalase, and glutathi-one peroxidase) and increased levels of oxidative damage markers (protein oxidation and 4-hydroxynonenal) in muscle tissue [109]. " Do rapid increases in oxidative stress, decreased activity of antioxidant enzymes (superoxide dismutase, catalase, and glutathione peroxidase) and increased levels of oxidative damage markers (protein oxidation and 4-hydroxynonenal) in muscle tissue lead to muscle atrophy? I recommend removing "muscle atrophy" from this sentence or changing the sentence. This sentence lists processes occurring at the cell level and between them "muscle atrophy".
R: The word muscle atrophy was removed. 389-393.
- Line 363. Did you mean the oxidative capacity of mitochondria? Please indicate whose oxidative capacity exactly.
R: It was clarified. Line 413
- Line 367. I recommend removing the abbreviation "(mRNA)".
R: All inappropriate abbreviations were removed throughout the text.
- Line 385. Please indicate what oxidative capacity increased as a result of training. Did the oxidative capacity of muscles increase?
R: It was clarified. Line 435.
- Line 387. I recommend removing the abbreviation "(NCAM)".
R: All inappropriate abbreviations were removed throughout the text.
- Line 400-401. "cytochrome c oxidase II and IV activity, and mitochondrial content) " What did you mean? Cytochrome c oxidase is complex IV of the respiratory chain. What does II mean? What mitochondrial protein contents were increased? Please edit this sentence
R: It was corrected. Lines 450-451.
- Line 400. Please write the same way - "cytochrome c oxidase"- Line 400, "cytochrome C oxidase"- Line 71.
R: It was corrected. Lines 450-451
- Line 403. Either "brain-derived neurotrophic factor" or "(BDNF)" should be removed. I recommend removing the abbreviation "(DCX)".
R: All inappropriate abbreviations were removed throughout the text.
- Line 404. I recommend removing the abbreviation "(VDAC)".
R: All inappropriate abbreviations were removed throughout the text.
- Line 405. Please clarify, did you observe an increase in protein content or expression of SOD2?
R: It was corrected. Line 454.
- Line 407. Either "high-intensity interval training" or "(HIIT)" should be removed.
- All inappropriate abbreviations were removed throughout the text.
- Line 415. The abbreviation "VDAC" with its explanation should be added to the description of Figure 3.
R: It was added. Lines 471-472
Summary
- Line 474. Either "treatment-resistant depression high-intensity interval training" or "(TRD)" should be removed.
R: All inappropriate abbreviations were removed throughout the text.
- Line 475. I recommend removing the abbreviation "(PBMC)".
R: All inappropriate abbreviations were removed throughout the text.
- Line 480-481. Either "treatment- resistant depression high-intensity interval training" or "(TRD)" should be removed.
R: All inappropriate abbreviations were removed throughout the text.
Abbreviations should be expanded in the abstract, figure captions, and once in the text of the article. They do not have to be cited and expanded in each chapter of the article.
R: All inappropriate abbreviations were removed throughout the text.
References
Please check that the references are formatted according to the formatting rules. Lines 653, 705, 715, 724, 737, 764, 772, 839, 847, etc. Somewhere the journal titles are given in full, somewhere - in abbreviated form. The journal titles and year are not highlighted (For example, Lines 563, 565, etc).
Please check that the references are formatted in accordance with the journal rules:
R: All references were corrected according to the journal format.

Round 2
Reviewer 1 Report
Comments and Suggestions for Authors
The authors answered all the questions. No more concerns.